# Studies on the Electro-Impulse De-Icing System of Aircraft

**Xingliang Jiang * and Yangyang Wang ***

State Key Laboratory of Power Transmission Equipment & System Security and New Technology, Chongqing University, Chongqing 400044, China

* Correspondence: Xingliangjiang@cqu.edu.cn (X.J.); 20161101022@cqu.edu.cn (Y.W.)

**Abstract:** In order to solve the accidents caused by aircraft icing, electro-impulse de-icing technology was studied through numerical simulation and experimental verification. In addition, this paper analyzed in detail the influence of the number, placement arrangement, and starting time of pulse coils on the de-icing effect, which plays a guidance role in the design and installation of the subsequent electro-impulse de-icing system. In an artificial climate chamber, the new de-icing criteria were obtained by tensile test, and the platform for the electro-impulse de-icing system was built. Replacing the skin with an aluminum plate, an electro-impulse de-icing system with a single coil was used. A three-dimensional skin-ice layer model was established by using Solidworks software. The finite element method was adapted. Through comparison between the de-icing prediction results and the test results in the natural environment, it was proven that the calculation process of de-icing prediction was correct, which laid a theoretical foundation for the selection of the number, placement arrangement, and starting time of the pulse coils. Finally, in this paper, by choosing the leading edge of NACA0012 wing as the research object, the influence of the number, placement arrangement, and starting time of pulse coils on the de-icing effect was analyzed. The results show that to get a better de-icing effect, the electro-impulse de-icing system with two impulse coils should be selected. The two coils were installed in the central position of the top and bottom surfaces of the leading edge, respectively. In addition, one of the impulse coils started working 1200 μs later than the other one.

**Keywords:** aircraft; wing; system; ice; icing; deicing; ice mechanics; electro-impulse; coils; structural dynamics

## 1. Introduction

Since the earliest days of aeronautics, icing was found to be a crucial problem for aircraft flight. In-flight ice accretion occurs on the leading edge of an aircraft wing and usually covers only 2% of the wing chord, with the thickness of the ice layer being about a few centimeters. However, even an ice layer of a few centimeters thickness at the key parts of the aircraft is enough to cause flow separation and destroy lift, increase drag and reduce the maximum lifting capability, affect the control surface effectiveness, and in some cases decrease engine performance and stability [1–3]. Therefore, people have been working on aircraft anti-icing methods for many years [4–6]. The present ice protection methods are hot bleed air, freezing point depressants, and electro-thermal resistance heating. However, they all have potential limitations, for example, pneumatic boots bonded to the ice prone surface are subject to corrosion and damage by external objects, therefore, they need to be replaced every two or three years. The freezing-point depressant systems achieve the purpose of aircraft anti-icing by releasing ethylene glycol through many orifices in the leading edge of an aircraft. There are two main hazards: on the one hand, it increases the weight of the aircraft; on the other hand, since ethylene glycol is a toxic

substance, release of this substance into the air can cause environmental pollution. The principle of electro-thermal systems is to use resistance pads to heat the aircraft to above the melting temperature of ice, and in general it requires dedicated generators, which results in significant cost.

The aforementioned methods cannot meet the performance requirements for new aircraft. Therefore, it is imperative to develop a safe and reliable de-icing system. A new method for electro-impulse de-icing has emerged.

Electro-impulse de-icing (EIDI) is one of the mechanical de-icing systems, and it assures the safety of aircrafts in an icing condition. It has major advantages, such as low energy, minimal maintenance, great reliability, and low cost and weight [7,8].

The basic circuit of the electro-impulse de-icing system is shown in Figure 1. The working principle is the pulse coils are connected to a high voltage capacitor by low resistance, low inductance cables. When the switch is turned on, the discharge of the capacitor through the impulse coils creates a rapidly forming and collapsing electro-magnetic field. According to Maxwell's law, we know that the time-dependent magnetic field induces eddy currents in the metal skin. Therefore, the instantaneous impulse force of several hundred pounds magnitude is obtained by the Lorentz force formula, but the duration is only a few hundred microseconds. A small amplitude, high acceleration movement of the skin acts to shatter, de-bond, and expel the ice.

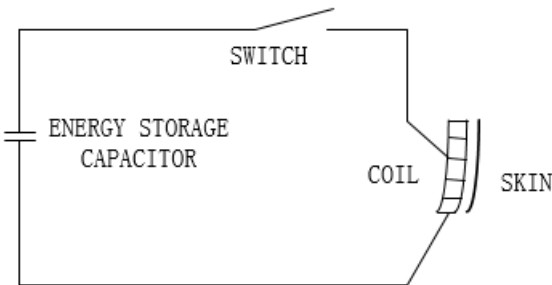

**Figure 1.** Electro-impulse system basic circuit.

Researchers [9–11] have studied the phenomenon of ice layer peeling on the surface of the skin under the action of a single coil system. The number of coils is greater than one when a real EIDI system is installed on the aircraft [12–14], so the position arrangement and start-up time of the multiple coils may affect the vibration condition to the effect of the ice removing effect. The de-icing criteria used in previous studies [15–17] are from the literature [18]. The adhesion between the ice layer and the skin is different under different icing conditions. Therefore, the conclusions of [18] are not suitable for all icing environments. Ice tensile strength and ice/aluminum adhesive shear strength used in [19] are 1.5 Mpa and 0.5 Mpa, respectively. However, this does not explain how the values are obtained. Therefore, the study of de-icing criteria is very important for the electro-impulse de-icing system.

All the experiments in this paper were obtained in an artificial climate chamber. Replacing the skin with an aluminum plate, the electro-impulse de-icing system with a single coil was used. The de-icing range obtained by experiments after each pulse was compared with the calculated result, and it was proven that the calculation process for de-icing prediction was correct. In addition, in the same icing environment, the adhesion between the ice layer and the skin was measured by a tensile test, and the new de-icing criteria was obtained. Finally, a three-dimensional NACA0012 wing-ice layer model was established; the effect of the number, placement arrangement, and the start-up time of pulse coils on the de-icing ratio was analyzed, which provided a reference for the design and installation of the subsequent electro-impulse de-icing system.

## 2. System Composition and Structural Dynamic Studies

The structural dynamics equation is expressed as [20]

$$[M]\ddot{U} + [C]\dot{U} + [K]U = F, \tag{1}$$

where [M] is the overall quality matrix, [C] is the overall damping matrix, [K] is the overall stiffness matrix, F is the impulse force, and U is the node displacement, $\ddot{U}$ is the node accelerated velocity, $\dot{U}$ is the node velocity.

In the continuous linear elastic body vibration, the relationship between normal stress, shear stress and displacement and external force is described. The expression for the stress can be written as

$$[\sigma] = [D][B]\{U\}^e. \tag{2}$$

Substituting the node displacement into Equation (2), $\sigma_x, \sigma_y, \sigma_z, \tau_{xy}, \tau_{yz}, \tau_{xz}$ for each node can be obtained.

The characteristic equation for the stress state is solved by the theory of one-dimensional cubic equation. Therefore, the practical calculation formulae for the principal stress can be written as

$$\sigma_1 = \frac{I_1}{3} + 2\sqrt{\frac{-P}{3}}\cos\frac{\theta}{3},$$

$$\sigma_2 = \frac{I_1}{3} - \sqrt{\frac{-P}{3}}\left(\cos\frac{\theta}{3} - \sqrt{3}\sin\frac{\theta}{3}\right),$$

$$\sigma_3 = \frac{I_1}{3} - \sqrt{\frac{-P}{3}}\left(\cos\frac{\theta}{3} + \sqrt{3}\sin\frac{\theta}{3}\right),$$

where $P = \frac{3I_2 - I_1^2}{3}$; $q = \frac{9I_1 I_2 - 2I_1^3 - 27I_3}{27}$; $\theta = \arccos\left[-\frac{q}{2}\left(-\frac{p^3}{27}\right)^{-\frac{1}{2}}\right](0 < \theta < \pi)$.

From the formulae above, $\sigma_1, \sigma_2, \sigma_3$ can be obtained. Substituting $\sigma_1, \sigma_2, \sigma_3$ into Equation (3), the equivalent stress can be written as

$$\sigma = \sqrt{\frac{1}{2}\left[(\sigma_1 - \sigma_2)^2 + (\sigma_2 - \sigma_3)^2 + (\sigma_3 - \sigma_1)^2\right]^2}. \tag{3}$$

## 3. Calculation Method Verification

### 3.1. Geometric Model

In this paper, an aluminum plate was used as the research object, and the material was the same as the wing skin. The research of electro-impulse de-icing systems mainly includes three parts: electrodynamics research, structural dynamics research, and de-icing prediction. The electro-dynamics studies of this EIDI included the calculation of impulse current, the study of the magnetic field behavior, and the calculation of the impulse force of each grid point on the surface of the aluminum plate, which provided the basis for structural dynamic research. Structural dynamics studies include stress calculations, which provide the basis for de-icing prediction. The model was established by using Solidworks, with a length of 420 mm, width of 420 mm, and thickness of 1.5 mm. The pulse coil was located on the lower surface of the aluminum plate, and the gap between the impulse coil and the skin was 1 mm. It had an inner diameter of 20 mm and an outer diameter of 128 mm. The material characteristics are shown in Table 1.

The structural dynamics associated with electro-impulse de-icing have proven to be a difficult and challenging problem. The structural dynamics study is mainly to calculate the equivalent stress between the ice layer and aluminum plate under the action of pulse force. If the total pulse force is

applied to the aluminum plate, the calculation accuracy will be affected. Therefore, the aluminum plate was divided into different regions in the radial direction. Then, the impulse pressure at different radial points and different times was placed on the corresponding regions on the surface of the aluminum plate. This greatly improved the accuracy of the de-icing prediction. It is assumed that the ice layer with 4 mm thickness was evenly covered on the aluminum plate with 2.0 mm thickness. The material properties are shown in Table 2.

**Table 1.** Material characteristics.

| Material | Relative Permeability | Conductivity (S/m) | Density (Kg/m$^3$) |
|---|---|---|---|
| Aluminum plate | 1.000021 | $1.74 \times 10^7$ | 2780 |
| Impulse coil | 0.999991 | $5.8 \times 10^7$ | 8933 |

**Table 2.** Material properties.

| Material | E/GPa | v | $\rho/(kg \cdot m^3)$ |
|---|---|---|---|
| Aluminum | 7.1 | 0.33 | 2700 |
| Ice | 5.5 | 0.3 | 897 |

The size of the meshing affects the calculation results of structural dynamics. In this paper, the modal analysis of the aluminum plate was calculated by the finite element method. The calculated result of first-order natural frequency was 46.5 Hz. Considering the boundary conditions and bending vibration of the aluminum plate, the aluminum plate was regarded as a free Bernoulli–Euer beam model at both ends, and its natural frequency theoretical calculation formula [17,18] is as follows:

$$f_n = \frac{\pi}{2}\left(n+\frac{1}{2}\right)^2 \sqrt{\frac{EI}{\rho Al^4}} \ , \ n = 1, 2, \cdots , \tag{4}$$

where $l$ is the length of the aluminum plate, $A$ is the cross-sectional area, $\rho$ is the density, $I$ is the moment of inertia of the section, and $E$ is the elasticity modulus.

The first-order natural frequency obtained by the finite element method was 46.5 Hz, and the theoretical calculation was 44.5 Hz. The relative error between the two was very small, only 4.3%. It indicates that the meshing sizes above were feasible, which provided a basis for the correct structural dynamics calculation.

*3.2. De-Icing Criteria*

The key factor of de-icing prediction research is the de-icing criteria. According to [9], there are two main de-icing criteria. One is to consider only a single strength factor at the ice-skin interface, and the other is to consider the tensile and shear stresses at the ice-skin interface. However, no matter which kind of the de-icing criteria is used, the equivalent stress and the shear stress are both determined values ($\sigma_U$ = 1.44 MPa, $\tau_U$ = 0.4 MPa). This is contrary to the fact that the equivalent stress and the shear stress between the ice layer and aluminum plate is different under different icing environments.

In this paper, the equivalent stress and the shear stress between the ice and aluminum plate was measured by a tensile test. Figures 2 and 3 show the icing box and tension test device, respectively. The icing test was carried out in an artificial climate chamber. As shown in Figure 4, the height and the diameter of the artificial climate chamber were 11.6 m and 7.8 m, respectively. The wind speed in the built-in wind tunnel can be up to 10 m/s. The sprinkler system consisted of 14 high-pressure water mist nozzles.

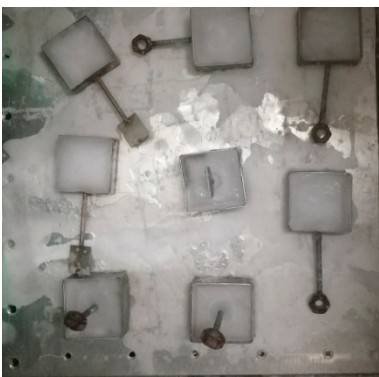

**Figure 2.** Icing box.

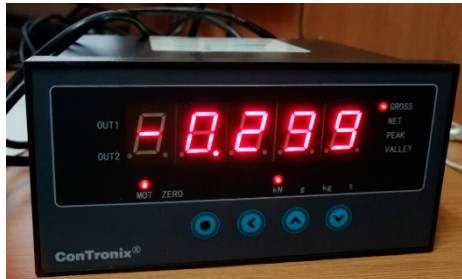

**Figure 3.** Tension sensor.

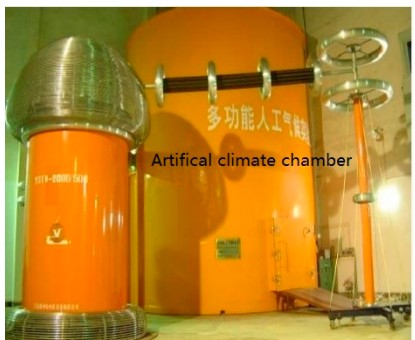

**Figure 4.** Artificial icing test.

The temperature curve during whole test was −20 degrees. Note that if a large amount of water is to be poured into the icing box at first, the water easily overflows. This causes icing around the outer surface of the icing box, resulting in inaccurate measurements. Therefore, a better option is to pour a little water into the icing box at first until the water in the icing box freezes completely, and repeat this procedure until the icing box is full of ice. In the natural icing environment, the outer surface of the icing box can also freeze. Before the tensile test, the icing around the icing box should be carefully removed so that the measurement result is accurate.

In addition, to improve measurement accuracy, the aluminum plate should be polished with gauze to remove the oxide film and rust on its surface. During the measurement process, the operating speed was controlled so that it accurately read the sensor readings. The results for the adhesion between the ice and the aluminum plate are shown in Table 3.

The tangential force is produced by electro-impulse de-icing system is small, therefore, this paper only needed to consider that the ice layer falls off when the tensile stress between the ice and aluminum

plate is greater than the maximum tensile stress. As shown in Table 3, the maximum tensile stress $\sigma_{max}$ between the ice and aluminum plate is

$$\sigma_{max} = \frac{F_{zave}}{S} = 0.15\ \text{MPa},$$

where $F_{zave}$ is the average value of the adhesion and S is the area of the ice box, 20.25 cm$^2$.

**Table 3.** The adhesion between the ice and the aluminum plate.

| Test Object | Equivalent Stress (kN) |
|:---:|:---:|
| (1) | 0.323 |
| (2) | 0.325 |
| (3) | 0.274 |
| (4) | 0.299 |
| (5) | 0.289 |
| (6) | 0.252 |
| (7) | 0.310 |

The maximum stress $\sigma_{max}$ between the ice and aluminum plate obtained in this paper and the result obtained by [21] are not much different. Therefore, it is indicative that the maximum stress $\sigma_{max}$ measured by tensile test is correct.

### 3.3. Result Analysis

#### 3.3.1. Analysis of Experimental Results

The icing test was carried out in natural icing station. The icing results of aluminum plates are shown in Figure 5. Icing thickness was 4 mm, as shown in Figure 6.

The circuit parameters are: voltage: U0 = 780 V; charging capacitor: C = 990 μf; inductance: L = 19 μH, $L_w$ = 5 μH. The de-icing results of skin are shown in Figure 7.

It can be seen from Figure 7 that the de-icing rate reached 73.6% after three-pulse. It is indicative that the electro-impulse de-icing system was feasible.

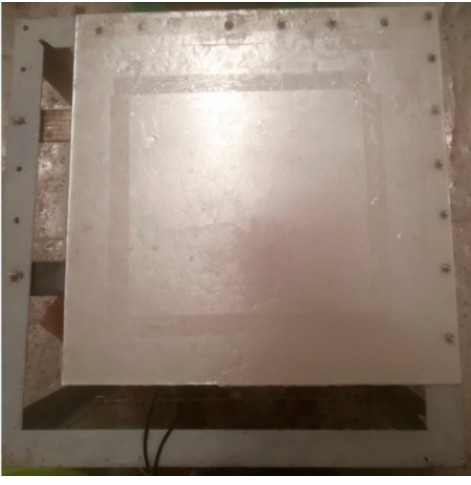

**Figure 5.** Icing results of the aluminum plate.

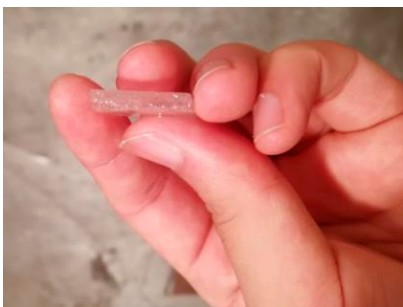

**Figure 6.** Icing thickness.

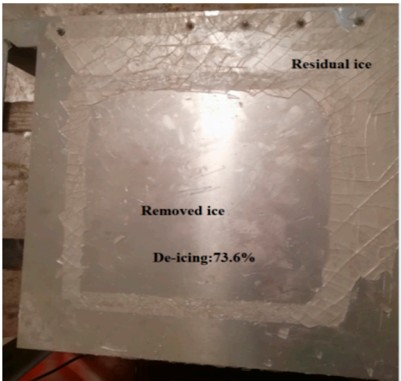

**Figure 7.** Ice layer shedding process.

### 3.3.2. Analysis of Calculation Results

A three-dimensional impulse coil-aluminum plate model was established by using Solidworks software. The importing of the model into the Maxwell software is shown in Figure 8a. An external circuit shown in Figure 8b was applied at the cross section of the impulse coil.

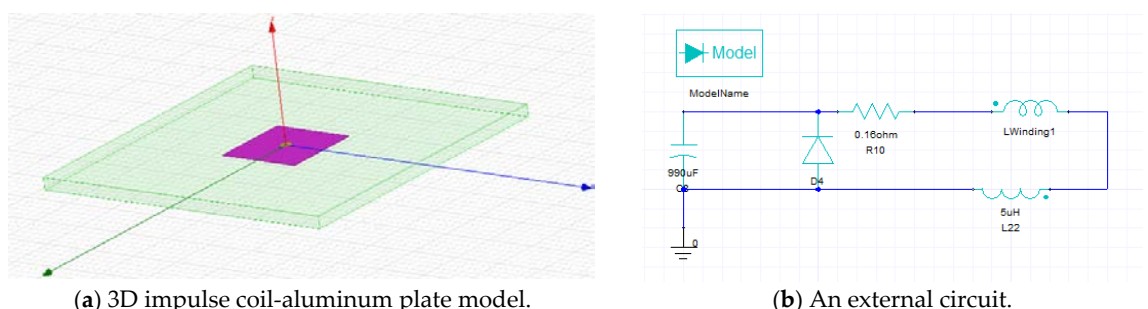

(**a**) 3D impulse coil-aluminum plate model.　　　　　(**b**) An external circuit.

**Figure 8.** Electro-dynamic simulation model.

Under the action of the instantaneous impulse force F, the equivalent stress between the skin and the ice layer can be obtained by Equations (1)–(3). The de-icing range on the surface of skin was found by using the new de-icing criteria. In order to improve the calculation result accuracy of de-icing prediction, the structural dynamic response was obtained by using the finite element method. The force at different times in different positions of the skin was applied to the corresponding position, as shown in Figure 9. The tangential magnetic field $B_r$ and the eddy current density $J_{eddy}$ at different positions and at different times of the aluminum plate were obtained by simulating. Therefore, the instantaneous impulse force can be obtained by Equation (5), as shown in Figure 10.

$$\vec{F} = \int_V \vec{J}_{eddy} \times \vec{B}dv, \tag{5}$$

where F is the impulse force, $J_{eddy}$ is the eddy current density, and B is the magnetic field.

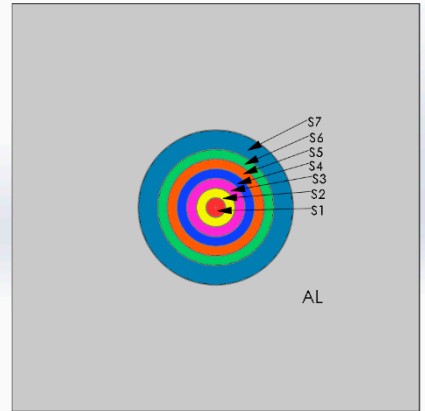

**Figure 9.** Excitation and loading area.

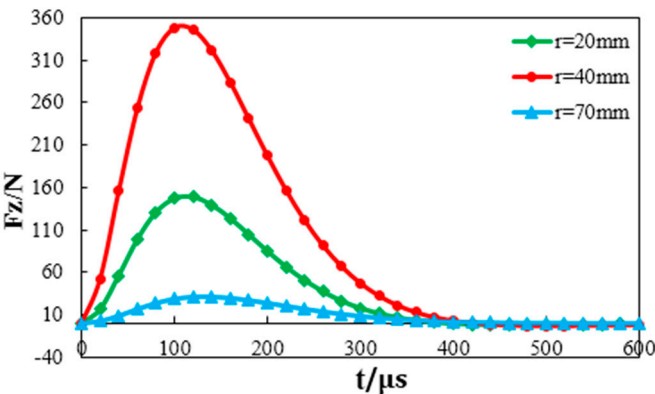

**Figure 10.** Relationship between force and time.

As shown in Figure 10, the distribution of force increased first and then decreased along the radial distance. The time required to reach the peak pressure was 100 μs, which meets the requirements of narrow pulse width.

According to the process of calculation, the equivalent stress was obtained by the finite element method, as shown in Figures 11 and 12.

The de-icing ratio obtained by the experiment was 73.6% after three pulses, and the de-icing ratio obtained by calculation reached 72.1% after three pulses. Although the de-icing ratio was very close, there was a difference in the de-icing range. The experimental results show that the lower and left side of the ice layer on the surface of the aluminum plate were completely detached, and the calculation results did not fall off. This was due to the effect of the external factors, such as wind speed. During the experiment, the ice on the top of the skin was easily able to fall off under the action of external wind speed. This factor was not considered in the calculation process of de-icing prediction.

In summary, the de-icing ratio obtained by the experiment was 73.6% after three pulses, and the de-icing ratio obtained by calculation reached 72.1% of the de-icing prediction after three pulses. In the range of errors permitted, the comparison between the experimental results and the predicted results indicated that the calculation process of de-icing prediction was correct.

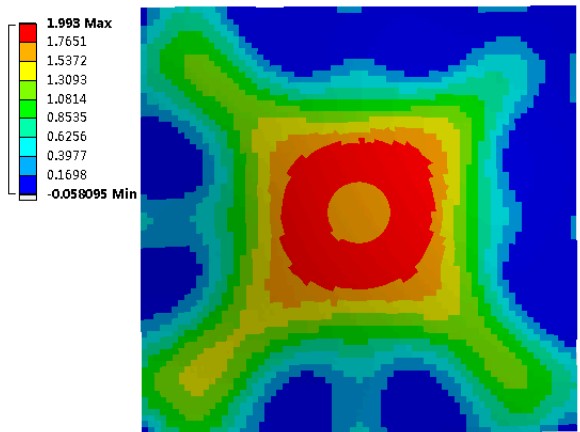

**Figure 11.** Equivalent stress.

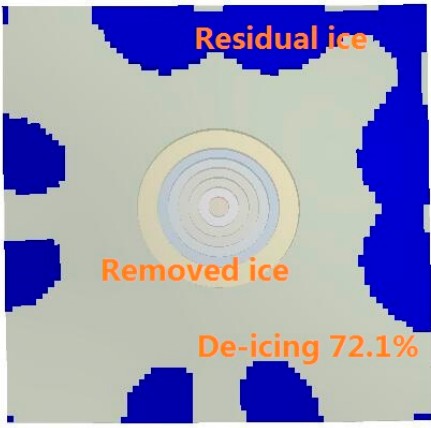

**Figure 12.** De-icing prediction results.

### 3.3.3. The Effect of Ice Thickness on the De-Icing Ratio

To study the changes of de-icing ratio with ice thickness while maintaining all other parameters unchanged, we assumed that the thicknesses of ice is 1 mm, 6 mm, 8 mm, and 10 mm, respectively. The de-icing ratios obtained by calculation are shown in Table 4.

**Table 4.** Effect of ice thickness on de-icing ratio.

| Ice Thickness (mm) | 1 | 6 | 8 | 10 |
|---|---|---|---|---|
| De-icing ratio | 80.7% | 71.04% | 53.44% | 10.42% |

It can be seen from Table 4 that the thicker the ice is, the smaller the de-icing ratio will be while maintaining all other parameters unchanged.

### 4. Example

The number of pulse coils was greater than one when a real EIDI system was installed on the aircraft, so the number, position arrangement, and start-up time of the multiple coils may have affected the vibration condition to the effect of the ice removing effect. The NACA0012 wing with a chord length of 1000 mm was taken as the research object in this paper. Since the wing was covered with ice mostly in the leading edge part [22–24], it was only necessary to analyze the de-icing ratio of the leading edge of the wing. A three-dimensional wing leading edge-ice layer model was established by using Solidworks, as shown in Figure 13. Its length along the chord was 300 mm, the length along the

exhibition was 600 mm, and the span of one bay of the leading edge was 300 mm. The thickness of the leading edge was 1.5 mm and the thickness of the ice layer was 1.0 mm.

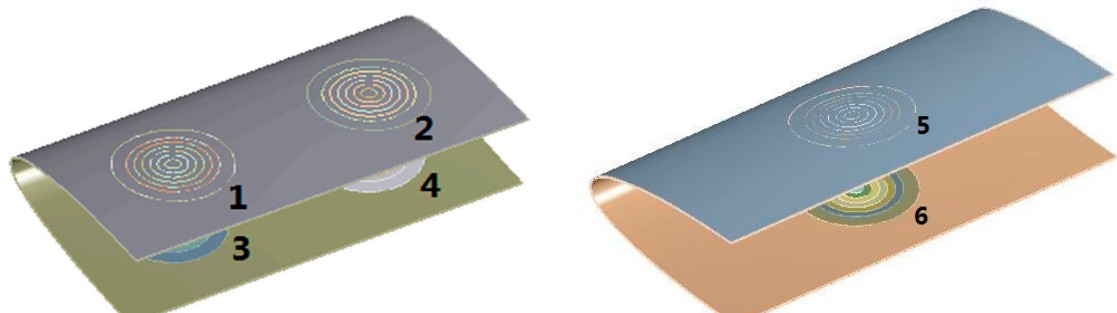

(**a**) Electro-impulse de-icing system with four coils.　　　(**b**) Electro-impulse de-icing system with two coils.

**Figure 13.** Position arrangement of multiple pulse coils.

### 4.1. Position Arrangement of the Multiple Coils

In order to study the influence of multiple coils on the de-icing ratio, the single coil simulation method was extended to a multiple coil simulation. Regarding the possibilities for weight reduction of EIDI systems, it was also important to determine the minimum number of coils, that is, whether it is necessary to place a coil in every bay of the leading edge. Because the established model was symmetrical, it only needed to select two or four pulse coils. The position arrangements of the four and two pulse coils are as shown in Figure 13a,b, respectively. The structure dynamics simulation of the four coil system was carried out. If only two of four coils were used, it was divided into three kinds of modes, which are respectively Mode 1, Mode 2 and Mode 3, as shown in Table 5. Mode 4 is the two coil system.

**Table 5.** Four kinds of two coils.

| Mode | (1) | (2) | (3) | (4) |
|---|---|---|---|---|
| Position arrangement | 1, 2 or 3, 4 | 1, 3 or 2, 4 | 1, 4 or 2, 3 | 5, 6 |

The de-icing results of simultaneous vibration of the four coils are shown in Figure 14. For the electro-impulse de-icing system of the four coils, the de-icing results of the simultaneous vibration are shown in Figures 15–17, in which only two of four coils were used. The de-icing results of simultaneous vibration of the two coils are shown in Figure 18.

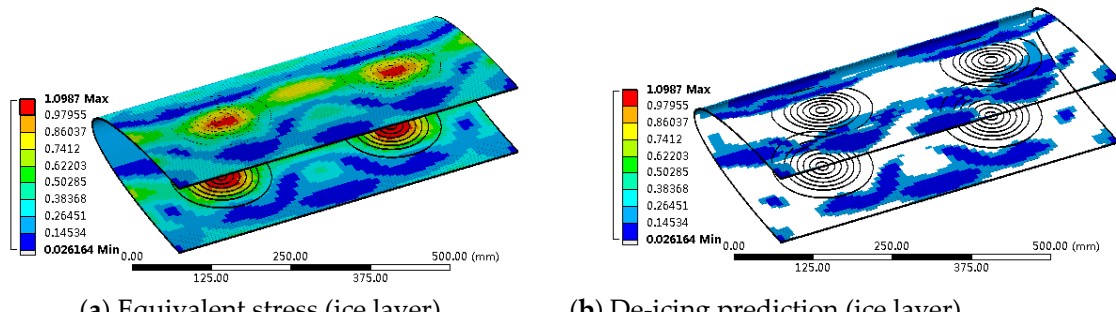

(**a**) Equivalent stress (ice layer).　　　　(**b**) De-icing prediction (ice layer).

**Figure 14.** Electro-impulse de-icing system of four coils.

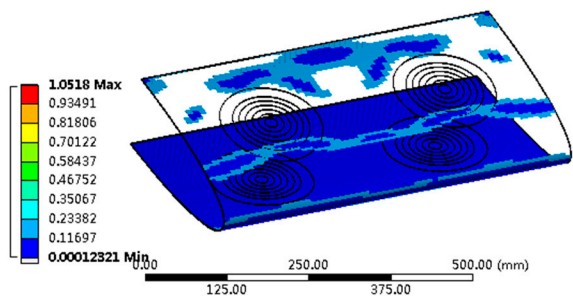

**Figure 15.** 1, 2 or 3, 4 of four coils are used (Mode 1).

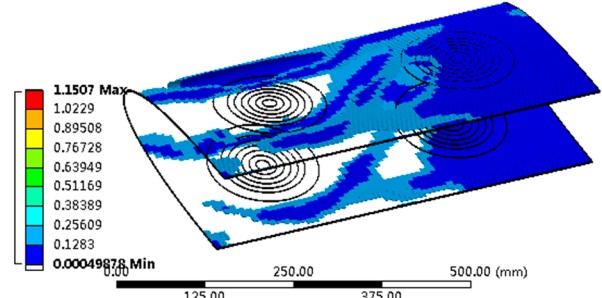

**Figure 16.** 1, 3 or 2, 4 of four coils are used (Mode 2).

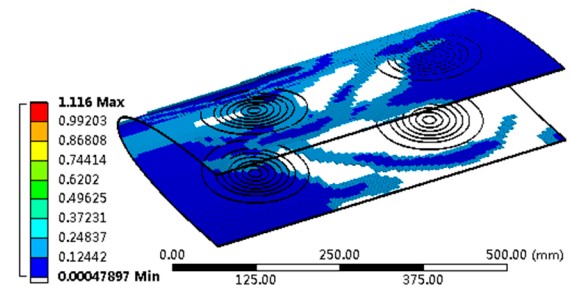

**Figure 17.** 1, 4 or 2, 3 of four coils are used (Mode 3).

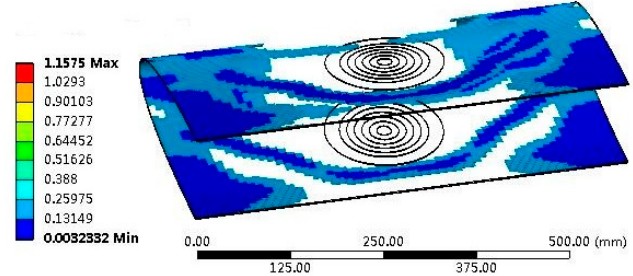

**Figure 18.** Electro-impulse de-icing (EIDI) system of two coils (Mode 4).

It can be seen from Figures 14–18 that the de-icing ratios varied with the number or the position arrangement of pulse coils. In addition, the ice layer fell off within a certain distance from the central position of the pulse coil. This was because the magnetic field was stronger in this range. The stronger the magnetic field, the bigger its magnetic force, thus, the ice layer was easy to fall off. The de-icing ratios of the several models above of the electro-impulse de-icing system are shown in Table 6.

It can be seen from Table 6 that for the electro-impulse de-icing system with two pulse coils, the de-icing ratio of Modes 1, 2, and 3 were only 36.19%, 38.35%, and 41.43%, respectively. The de-icing ratio of Mode 4 was 47.22%, which was the maximum. The de-icing ratio of the electro-impulse de-icing system with four coils was 69.83%. According to the experimental results, the ice layer at the

boundary of the skin would also fall off under the influence of external factors. Therefore, from the perspective of not affecting aircraft flight and low energy consumption, the position arrangement of Mode 4 is optimal for the research objects mentioned above.

**Table 6.** De-icing ratio comparison.

|  | Electro-Impulse De-Icing System with Four Coils | Electro-Impulse De-Icing System with Two Coils | | | |
|---|---|---|---|---|---|
|  |  | Mode 1 | Mode 2 | Mode 3 | Mode 4 |
| De-icing ratio | 69.83% | 36.19% | 38.35% | 41.43% | 47.22% |

### 4.2. The Effect of Start-Up Time of Pulse Coil

In the case of Mode 4, the starting times of the pulse coils 5 and 6 were different, and the de-icing ratios obtained by the calculation were different. Take as an example the system with two pulses, pulse coil 5 started to work at 0 μs, and pulse coil 6 started to vibrate from 0 μs, 100 μs, 600 μs, and 1200 μs, respectively. The vibration de-icing result at 0 μs was the same as that of Mode 4. The results of the ice layer detachment of pulse coil 6 starting to work from 100 μs is shown in Figure 19b. Figure 19c shows the pressure waveform at the coil radial direction r = 30 mm. The results of the ice layer detachment of pulse coil 6 starting to work from 600 μs and 1200 μs are shown in Figures 20 and 21, respectively.

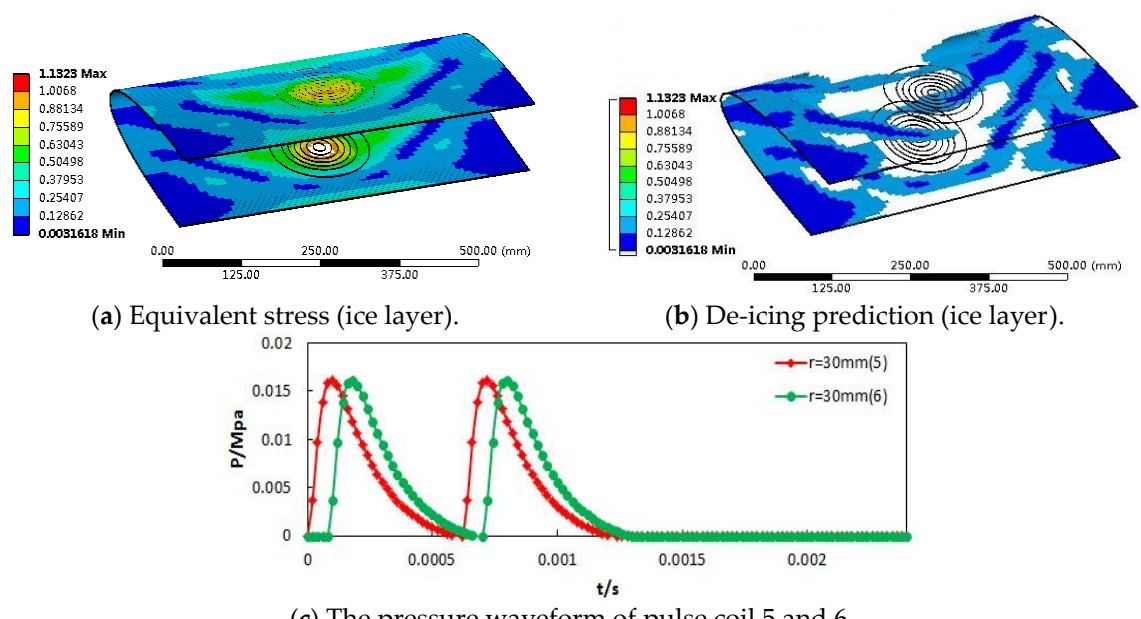

(**a**) Equivalent stress (ice layer).　　　　　　　(**b**) De-icing prediction (ice layer).

(**c**) The pressure waveform of pulse coil 5 and 6.

**Figure 19.** De-icing results of pulse coil 6 starting 100 μs later than coil 5.

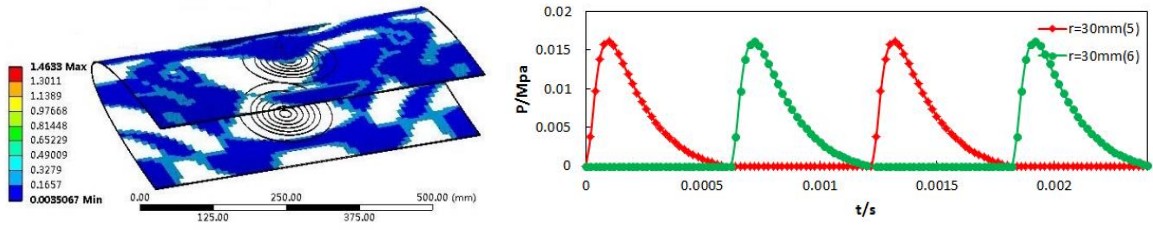

(**a**) De-icing prediction (ice layer).　　　　　(**b**) The pressure waveform of pulse coil 5 and 6.

**Figure 20.** De-icing results of pulse coil 6 starting 600 μs later than coil 5.

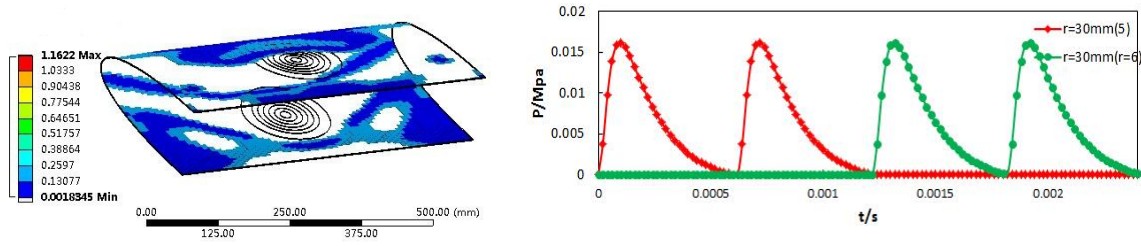

(**a**) De-icing prediction (ice layer).　　　　　(**b**) The pressure waveform of pulse coil 5 and 6.

**Figure 21.** De-icing results of pulse coil 6 starting 1200 μs later than coil 5.

It can be seen from Figure 19 that the de-icing ratios obtained by calculation were different under the different starting times of pulse coil 6. De-icing prediction results are shown in Table 7.

**Table 7.** De-icing ratios.

| Start Time Delay of Coil 6 with Respect of Start Time of Coil 5 | 0 μs | 100 μs | 600 μs | 1200 μs |
|---|---|---|---|---|
| De-icing ratio | 47.22% | 51.38% | 45.42% | 54.03% |

It can be seen from Table 7 that if pulse coil 6 started working at the same time as coil 5, or 600 μs later, the de-icing ratios obtained by calculation were almost the same. This was because the interval between the continuous applied force at the same position of the wing was the same. However, if pulse coil 6 started working 1200 μs later than coil 5, the de-icing ratio was the largest, which served as a guiding function for the subsequent pulse coil vibration control.

## 5. Conclusions

(1) In an artificial climate chamber, the maximum adhesion between ice layer and aluminum plate was 0.15 Mpa obtained by the tensile test. Since the tangential force generated by the electro-impulse de-icing system was very small, as long as the equivalent stress between the ice layer and the aluminum plate was greater than 0.15 Mpa, the ice layer fell off, which laid a foundation for the simulation calculation of the subsequent electro-impulse de-icing system.

(2) We replaced the skin with an aluminum plate and built a platform of the electro-impulse de-icing system in an artificial climate chamber. The range of de-icing of the skin surface obtained by the experiments was recorded under different pulse times. A three-dimensional aluminum plate-ice layer model was established by using Solidworks. The results obtained by the three-step calculation of electrodynamics, structural dynamics, and de-icing prediction were compared with the experiment results of the electro-impulse de-icing system. It was indicative that the calculation process of de-icing prediction obtained by this paper was correct.

(3) Taking the leading edge of the NACA0012 wing with a chord length of 1 m as the research object, its length along the chord was 300 mm, the length along the exhibition was 600 mm, and the span of one bay of the leading edge was 300 mm. The leading edge had a thickness of 2.0 mm and the ice thickness was 4.0 mm. The de-icing ratio of the four pulse coil system was 69.83%. Under the action of the two pulse coil system, the de-icing ratio of Mode 1, 2, and 3 was lower, and the de-icing ratio of Mode 4 reached 47.22%. From the perspective of low energy consumption and without affecting the safe flight of the aircraft, the aforementioned structure was best to select the electro-impulse de-icing system with two pulse coils.

(4) With regard to the electro-impulse de-icing system with two-pulse coils, the starting time of the pulse coil was different, and the de-icing effect was different. If pulse coil 6 started working at the same time as coil 5 or 600 μs later, the de-icing ratio was almost the same. This was because the interval between the continuous applied pressure at the same position of the wing was the

same. However, if pulse coil 6 started working 1200 μs later than coil 5, the de-icing ratio was the largest, which served as a guiding function for the subsequent pulse coil vibration control.

**Author Contributions:** Conceptualization, X.J. and Y.W.; methodology, Y.W.; software, Y.W.; writing—original draft preparation, Y.W.; writing—review and editing, X.J. and Y.W.

**Funding:** (1) This research was funded by the National Natural Science Foundation of China, grant number 51637002. (2) This research was funded by State Grid Science and Technology Project: Research on numerical prediction technology of grid icing in micro-topography and micro-meteorological areas, grant number 521999180006. (3) This research was funded by the Fundamental Research Funds for the Central Universities, grant number 2019CDXYDQ0010.

**Acknowledgments:** The presented work is part of the electro-impulse de-icing system for aircraft, which is funded by State Grid Corporation of China.

**Conflicts of Interest:** The authors declare no conflict of interest.

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
