# Peer review of "Studies on the Electro-Impulse De-Icing System of Aircraft"

_aerospace, doi:10.3390/aerospace6060067_

Round 1

Reviewer 1 Report

The subject of the paper is interesting.

Main points: The article presents electro-impulse de-icing tests and proposes modeling tracks for predicting plate or NACA profile defrosting capability.

State of the art: The state of the art is incomplete. A number of recent papers on the subject are missing. For example, the paper does not cite recent work in the electro-impulse field of Braunchweig University and DLR as
Endres, M., Sommerwerk, H., Mendig, C., Sinapius, M., & Horst, P. (2017). Experimental study of two electro-mechanical de-icing systems applied on a wing section tested in an icing wind tunnel. CEAS Aeronautical Journal, 8 (3), 429-439
Sommerwerk, H., Horst, P., & Bansmer, S. (2016). Studies on electronic impulse de-icing of a leading edge structure in an icing wind tunnel. In 8th AIAA Atmospheric and Space Environments Conference (3441).
On the de-icing criteria in electro-mechanical technology, the works of Palacios, Budinger, Benanni are not mentioned.

Quality of writing: we can note a number of typographical errors as
l 56 and others: the references sometimes have [], sometimes not
l 85: reference to a nonexistent equation (25)
etc.

Deicing mechanisms: it would be interesting to better explain the phenomena leading to deicing. It is not clear in the article whether the normal stress or shear criterion dominates. A diagram and an explanation of the sources of constraints (inertial, elastic deformation, ...) would be welcome. An order of magnitude calculation would justify the modeling and measurement choices. Nor does the article mention crack propagation, which is an important criterion for ice (brittle material).

The modeling approach: in equation (1), it is not explained how the transient F effort is obtained. No model of coil - induced current - structure interaction is presented. It is therefore not understood how the results of section 3.2.3 are obtained. The final article should have more explanations at this level.

Author Response

Response to the reviewer 1:

Thank you for your careful review of the paper manuscript.

Based on your comments on the paper, we have revised the draft in the following aspects (marked by Red Colour Font):

The subject of the paper is interesting.

Main points: The article presents electro-impulse de-icing tests and proposes modeling tracks for predicting plate or NACA profile defrosting capability.

(1)     State of the art: The state of the art is incomplete. A number of recent papers on the subject are missing. For example, the paper does not cite recent work in the electro-impulse field of Braunchweig University and DLR as Endres, M., Sommerwerk, H., Mendig, C., Sinapius, M., & Horst, P. (2017). Experimental study of two electro-mechanical de-icing systems applied on a wing section tested in an icing wind tunnel. CEAS Aeronautical Journal, 8 (3), 429-439.

Sommerwerk, H., Horst, P., & Bansmer, S. (2016). Studies on electronic impulse de-icing of a leading edge structure in an icing wind tunnel. In 8th AIAA Atmospheric and Space Environments Conference (3441).
On the de-icing criteria in electro-mechanical technology, the works of Palacios, Budinger, Benanni  are not mentioned.

Response 1: A number of recent papers on the subject have been added in this paper (Page 13).

On the de-icing criteria in electro-impulse de-icing technology, the works of Palacios, Budinger, Benanni have been added to in the paper.

The Ice tensile strength and Ice/aluminum adhesive shear strength used in Budinger’s paper are 1.5Mpa and o.5Mpa, respectively (Page 2). But the paper does not explain how the values are obtained. In addition, we found that Budinger and Palacios mainly study Piezoelectric Ice Protection System, which is different from electro-impulse de-icing.

No related literatures on de-icing criteria were found in Benanni’s articles.

(2)     Quality of writing: we can note a number of typographical errors asl 56 and others: the references sometimes have [], sometimes not l 85: reference to a nonexistent equation (25) etc.

Response 2: Reference types in the paper have been revised. We have changed the number of equation 25 (page 3).

(3) Deicing mechanisms: it would be interesting to better explain the phenomena leading to deicing. It is not clear in the article whether the normal stress or shear criterion dominates. A diagram and an explanation of the sources of constraints (inertial, elastic deformation, ...) would be welcome. An order of magnitude calculation would justify the modeling and measurement choices. Nor does the article mention crack propagation, which is an important criterion for ice (brittle material).

Response 3: The de-icing conditions of the leading edge of the wing were described in this paper (in Section4).

Since the different icing environments, the icing shapes in the leading edge of the wing are different, and usually covers only 2% of the wing chord. Considering all types of icing, a special icing shape of NACA0012 wing was analyzed in section 4 of this manuscript.

Since the shear stress between the ice and the skin measured by the tensile test is greater than the adhesion. However, the calculated tangential pulse force is smaller than the vertical pulse force. This paper considers that as long as the adhesion calculated by structural-dynamics studies between the ice and the skin is greater than the maximum adhesion calculated by the tensile test, the ice layer falls off.

the research content of this paper is only the preliminary work, we will study the effects of crack propagation in the later stage.

(4) The modeling approach: in equation (1), it is not explained how the transient F effort is obtained. No model of coil - induced current - structure interaction is presented. It is therefore not understood how the results of section 3.2.3 are obtained. The final article should have more explanations at this level.

Response 4: A three-dimensional coil-skin model has been established, and the solving process of transient force has described in detail in the section 3.2.3(Page 7).

We are grateful to you for your comments on the former draft and looking forward to your review of the revised paper.

Sincerely yours

Xingliang Jiang

Reviewer 2 Report

Review comments

Title: Experiment and Simulation of Electro-impulse De-icing process of Aircraft

Manuscript id: aerospace-476346

General comments:

The manuscript “Experiment and Simulation of Electro-impulse De-icing process of Aircraft” presents a de-icing prediction by calculating the von Mises stress over the ice-skin interface induced by an electro-impulse device.

This is an interesting work on the prediction of icing mitigation. However, there are a number of points of criticism which need to be addressed. I recommend major revision before the publication can be considered.

Detailed comments:

1.      Authors should have a proof reading and check grammar. In the manuscript there are several mistakes such as capital letter use and punctuation mistakes. For example, on line 27, change “Hot bleed air” to “hot bleed air”, “Freezing point depressants” to “freezing point depressants”, “Electro-thermal” to “electro-thermal”.

2.      On page 2 line 56; authors mentioned about “Documents 12-14”. “What are these documents?”

3.      On page 2 line 57; authors mentioned the conclusion of the “literature 15” is not suitable for all icing environments. However, the study in the literature 15 focuses on impact icing phenomena, which is more relevant to aircraft in-flight icing. Authors should explain why proposed “de-icing criteria” need to be improved.

4.      On page 3 line 85; change “?2?2, ?3” to “?2, ?2, ?3

5.      On page 3 line 85; mentioned equation number is wrong. There is no equation 25 in the manuscript.

6.      Figure 2 seems unnecessary. It can be removed.

7.      In section 3.2, authors mentioned about their test facility. But experimental conditions were not given. Authors should explain, how the ice formed on test samples, what is the temperature. Temperature being -40°C or -2°C would affect the ice structure.

8.      On page 5 table 3, authors show results for adhesion between ice and aluminum. Authors should mention how they performed this test and compare their results with the literature.

9.      For removal of the ice, authors only considered the adhesion between ice and the aluminum plate. Authors should comment on shear strength of ice. Factors such as temperature, ice thickness and type of ice would highly influence the shear yielding of ice structures.

10.   Authors mentioned (page 9) the ice falling of due to external wind and gravity. In my opinion, such external factors should be minimized when directly comparing experimental and computational results. This is one of the reason why experimental and computational results in this manuscript do not match well. De-iced areas of experiments and simulations are different for two impulses and three impulses.

11.   In figure 11 and 12 de-icing prediction after two impulses and three impulses were shown. After two impulses, it is shown that, there is small piece of ice removed on top right of the plate. However, for the results after three impulses, ice is remained at that corner. Authors should explain.

12.   Plots showing the de-icing prediction (fig 10b-12b, 14b-21b) are basically same figure with equivalent stress plots. Only difference is the color change of contours. Also the plots have too many writing on them, which makes them look crowded. The unnecessary information can be removed and plots can be simplified.

13.   Section 4 of this manuscript includes an example of electro-impulse de-icing prediction on a NACA 0012 airfoil. The conditions during the flight -such as wind speed, aerodynamic forces, temperature, ice formation- would be different than what we have on the ground. De-icing prediction on the ground cannot be simply applied to the inflight de-icing.

Author Response

Response to the reviewer 2:

Thank you for your careful review of the paper manuscript.

Based on your comments on the paper, we have revised the draft in the following aspects (marked by Red Colour Font):

The manuscript “Experiment and Simulation of Electro-impulse De-icing process of Aircraft” presents a de-icing prediction by calculating the von Mises stress over the ice-skin interface induced by an electro-impulse device.

This is an interesting work on the prediction of icing mitigation. However, there are a number of points of criticism which need to be addressed. I recommend major revision before the publication can be considered.

(1)    Authors should have a proof reading and check grammar. In the manuscript there are several mistakes such as capital letter use and punctuation mistakes. For example, on line 27, change “Hot bleed air” to “hot bleed air”, “Freezing point depressants” to “freezing point depressants”, “Electro-thermal” to “electro-thermal”.

Response 1: Following the suggestion of the reviewer, we have revised the language in the whole manuscript, which is marked by Red Colour Font.

(2) On page 2 line 56; authors mentioned about “Documents 12-14”. “What are these documents?”

Response 2: On page 2; “Documents 12-14” at where” Documents” is the meaning of “Reference”. We have modified it in the paper.

(3) On page 2 line 57; authors mentioned the conclusion of the “literature 15” is not suitable for all icing environments. However, the study in the literature 15 focuses on impact icing phenomena, which is more relevant to aircraft in-flight icing. Authors should explain why proposed “de-icing criteria” need to be improved.

Response 3: According to a large number of literatures at home and abroad, the adhesion stresses between ice and skin are different and change in large range in different icing environments. Because the adhesion stresses are different, the amplitudes and widths of pulse required by de-icing are different, that is, the power consumption of the electro-impulse de-icing system are different. From the perspective of optimal energy consumption, it is important to study the de-icing criteria.

(4) On page 3 line 85; change “σ2, σ3” to “σ2, σ2, σ3

Response 4: On page 3; “σ2, σ3” has been changed to “σ1, σ2, σ3”.

(5) On page 3 line 85; mentioned equation number is wrong. There is no equation 25 in the manuscript.

Response 5: On page 3; number of equation 25 is modified in the manuscript.

(6)  Figure 2 seems unnecessary. It can be removed.

Response 6: The Figure 2 has been removed.

(7)  In section 3.2, authors mentioned about their test facility. But experimental conditions were not given. Authors should explain, how the ice formed on test samples, what is the temperature. Temperature being -40°C or -2°C would affect the ice structure.

Response 7: We have given the experimental conditions and explained the formation process of ice in the icing box in section 3.2 (page 5).

(8) On page 5 table 3, authors show results for adhesion between ice and aluminum. Authors should mention how they performed this test and compare their results with the literature.

Response 8: We have described in detail the operational procedure of the tensile test. The results obtained in this paper are compared with the results obtained in reference 21, it is indicate that the maximum stress ?max  measured by tensile test is correct (page 6).

(9)  For removal of the ice, authors only considered the adhesion between ice and the aluminum plate. Authors should comment on shear strength of ice. Factors such as temperature, ice thickness and type of ice would highly influence the shear yielding of ice structures.

Response 9: Since the shear stress between the ice and the skin measured by the tensile test is greater than the adhesion. However, the calculated tangential pulse force is smaller than the vertical pulse force. This paper considers that as long as the adhesion calculated by structural-dynamics studies between the ice and the skin is greater than the maximum adhesion calculated by the tensile test, the ice layer falls off.

(10) Authors mentioned (page 9) the ice falling of due to external wind and gravity. In my opinion, such external factors should be minimized when directly comparing experimental and computational results. This is one of the reason why experimental and computational results in this manuscript do not match well. De-iced areas of experiments and simulations are different for two impulses and three impulses.

Response 10: Considering the effect of gravity on de-icing ratio, the de-icing prediction results for two impulses and three impulses have been recalculated (Page 8).

(11) In figure 11 and 12 de-icing prediction after two impulses and three impulses were shown. After two impulses, it is shown that, there is small piece of ice removed on top right of the plate. However, for the results after three impulses, ice is remained at that corner. Authors should explain.

Response 11: The new figure 13 shows that the ice on top right of the plate has been removed after three impulses.

(12)  Plots showing the de-icing prediction (fig 10b-12b, 14b-21b) are basically same figure with equivalent stress plots. Only difference is the color change of contours. Also the plots have too many writing on them, which makes them look crowded. The unnecessary information can be removed and plots can be simplified.

Reponse12: In order to make de-icing prediction diagrams (fig 10-12, 14-21) look crowded, we removed the equivalent stress diagrams of Figure 10-12 ,14-21 and only retained the de-icing prediction diagrams.

(13) Section 4 of this manuscript includes an example of electro-impulse de-icing prediction on a NACA 0012 airfoil. The conditions during the flight -such as wind speed, aerodynamic forces, temperature, ice formation- would be different than what we have on the ground. De-icing prediction on the ground cannot be simply applied to the inflight de-icing.

Response 13: In fact, the conditions during the flight-such as wind speed, aerodynamic forces, temperature and LWC are the main factors affecting the ice shape of the aircraft. Types of aircraft icing are mainly divided into rime ice, glaze ice and mixed ice. We have also done some research on icing shape prediction of aircraft. Considering all types of icing, a special icing shape of NACA0012 wing was analyzed in section 4 of this manuscript. the research content of this paper is only the preliminary work, we will study the effects of the conditions during the flight in the later stage.

We are grateful to you for your comments on the former draft and looking forward to your review of the revised paper.

Sincerely yours

Xingliang Jiang

Reviewer 3 Report

Interesting paper. This type of data (both experiment and analysis) supporting EIDI development is useful. The test is understandable but please try to improve the English as much as you can.

Some points you might consider adding:

1) How long and under what kind of environmental conditions did the ice (1mm) accreted for your experimental tests? (T,LWC,Velocity, humidity) ( do you have a measured value for density of ice for the tests?)(Section 3.3)

2)  What is the maximum measured displacement of the vertical plate? Some plot like Fig9 for Displacement vs time(wave propagation) Is there any correlation with displacement with ice de-icing%?

3) Note that tests were done on flat plate! leading edge is curved. Is there any correlation of ice removal with curvature?

4)How does ice removal change with ice thickness and aluminum skin thickness? Is there any non dimensional parameter one can correlate to?

Another point that can be connected with the introduction is that : Relationship of system weight and power consumption with ice removal capability?

Overall it is interesting work.

Regards

Author Response

Response to the reviewer 3:

Thank you for your careful review of the paper manuscript.

Based on your comments on the paper, we have revised the draft in the following aspects (marked by Red Colour Font):

Interesting paper. This type of data (both experiment and analysis) supporting EIDI development is useful. The test is understandable but please try to improve the English as much as you can

Following the suggestion of the reviewer, we have revised the language in the whole manuscript.

(1)     How long and under what kind of environmental conditions did the ice (1mm) accreted for your experimental tests? (T,LWC,Velocity, humidity) ( do you have a measured value for density of ice for the tests?)(Section 3.3)

Response1: The icing environmental conditions are given, as shown in Figure 2. the icing time (1mm) is 12h. The density of ice is about 0.9g/m^3 (Page 5).

(2)     What is the maximum measured displacement of the vertical plate? Some plot like Fig9 for Displacement vs time(wave propagation) Is there any correlation with displacement with ice de-icing%?

Response 2: Figure 9 (renumbering is Figure 10) show the relationship of pressure at different times in different positions of the skin varying with time (Page 8). We found that the de-icing ratio has nothing to do with displacement.

(3)     Note that tests were done on flat plate! leading edge is curved. Is there any correlation of ice removal with curvature?

Response 3: In fact, ice removal isn’t related to curvature. Because the pulse coil is actually installed through the structure, which presses against the pulse coil and cause it to conform to the curve of the airfoil.

(4) How does ice removal change with ice thickness and aluminum skin thickness? Is there any non dimensional parameter one can correlate to?

Another point that can be connected with the introduction is that : Relationship of system weight and power consumption with ice removal capability?

Response 4: While maintaining all other parameters unchanged, the thicker the ice is, the smaller the de-icing ratio will be. The relationship between ice removal and ice thickness has been added (Page 9).

Due to its proximity effect and skin effect, ice removal isn’t related to aluminum skin thickness.

Weight and power consumption of electro-impulse de-icing system are directly uncorrelated to de-icing capability. The weight mentioned in the introduction refers to the weight of the whole device of the electro-impulse de-icing system. Power consumption includes not only energy consumption for de-icing but also circuit losses.

We are grateful to you for your comments on the former draft and looking forward to your review of the revised paper.

Sincerely yours

Xingliang Jiang

Round 2

Reviewer 1 Report

The authors answer to the main critics done during the 1st review.

Author Response

Thank you for your careful review of the paper manuscript

Reviewer 2 Report

The revised manuscript is acceptable for publication

Author Response

(The authors gave the same response as above.)
